# Changes, Desire, Fear and Beliefs: Women’s Feelings and Perceptions About Dental Care During Pregnancy

**DOI:** 10.3390/ijerph22081211

**Published:** 2025-07-31

**Authors:** Natália Correia Fonseca Castro, Vânia Maria Godoy Pimenta Barroso, Henrique Cerva Melo, Camilla Aparecida Silva de Oliveira Lima, Rafaela Silveira Pinto, Lívia Guimarães Zina

**Affiliations:** Faculty of Dentistry, Universidade Federal de Minas Gerais, Belo Horizonte 31270-901, MG, Brazil; nataliacfc@ufmg.br (N.C.F.C.); vaniagodoy@hotmail.com (V.M.G.P.B.); henriquecm@ufmg.br (H.C.M.); camillaaparecida@ufmg.br (C.A.S.d.O.L.); rafaelasilveirapinto@ufmg.br (R.S.P.)

**Keywords:** pregnant women, oral health, perception, prenatal care, primary healthcare, Unified Health System

## Abstract

Oral health during pregnancy is essential for maternal and child well-being, as hormonal and physiological changes increase women’s susceptibility to oral diseases. Despite the recognized importance of prenatal dental care, adherence to dental services remains a challenge in the public health context. This study aimed to analyze oral health and the use of dental services during pregnancy through the perception of pregnant women. It represents the qualitative phase of a mixed-method study conducted with 25 pregnant women (with and without dental care) receiving prenatal care in the Brazilian Unified Health System (SUS). Participants were selected through saturation sampling, and data were collected via semi-structured interviews, followed by content analysis. The findings revealed four major themes: barriers and facilitators to dental care, changes during pregnancy and oral health. Discomfort from oral changes was a common concern. Barriers included misinformation, fear, cultural beliefs, and service organization. In contrast, facilitating factors were identified, such as care prioritization, support from healthcare teams, health education, and access through SUS. This study concludes that emotional, cultural, and contextual aspects shape the use of dental services during pregnancy. Access through SUS is perceived as an important facilitator, which simultaneously presents organizational weaknesses that need to be addressed.

## 1. Introduction

Pregnancy is a period of intense physiological and metabolic changes in a woman’s body, including hormonal, immunological, and nutritional alterations, which can directly affect the oral health of the pregnant woman [1]. Increased estrogen and progesterone levels during pregnancy heighten the gingival inflammatory response, causing peripheral vasodilation, pregnancy gingivitis, and deeper periodontal pockets [2]. Significant changes in oral microbiota promote the growth of periodontopathogenic bacteria such as Porphyromonas gingivalis and Prevotella intermedia, increasing the risk of severe periodontitis [3]. Other common manifestations include physiological xerostomia, which alters salivary pH and buffering capacity [4], as well as enamel erosion, often associated with vomiting and gastric reflux [1]. These changes create a favorable environment for the development of oral diseases such as pregnancy gingivitis, periodontitis and dental caries [5], making dental care an essential component of prenatal care. Prenatal dental care (PDC) consists of a set of preventive, educational, and therapeutic measures aimed at promoting oral health among pregnant women and preventing conditions that could compromise both maternal health and fetal development [6]. Integrating dental care into prenatal care aligns with the principle of comprehensive healthcare and should be prioritized within primary healthcare (PHC) settings to ensure safe and humanized care for pregnant women [7].

Beyond oral health promotion, PDC is relevant to preventing adverse obstetric outcomes. Studies have identified periodontal diseases as potential risk factors for obstetric complications such as preterm birth, low birth weight and intrauterine growth restriction [8,9]. These complications are associated with bacterial dissemination originating from the oral cavity, which can reach the placenta and amniotic fluid, triggering exacerbated inflammatory responses capable of inducing early uterine contractions and activating pathophysiological mechanisms involved in the induction of labor [10]. Evidence suggests that pregnant women with periodontitis are also at higher risk of developing preeclampsia due to chronic infection-induced systemic inflammation, which elevates cytokines like IL-6 and TNF-α, contributing to endothelial dysfunction and hypertension [11]. Similarly, chronic periodontitis has been associated with gestational diabetes, as inflammation disrupts glucose homeostasis and reduces insulin sensitivity [11]. However, these associations should be analyzed with caution, since methodological limitations in many studies preclude definitive conclusions regarding the direct impact of periodontal disease on pregnancy outcomes [12]. Notably, the burden of periodontitis-related complications appears higher in low-income countries like Brazil compared to high-income settings such as the United States and the United Kingdom [13].

Recognizing the importance of PDC, Brazil has progressively incorporated it into public health policies. Every pregnant woman receiving prenatal care in primary healthcare services should be referred for dental care [14]. Despite Brazil being one of the few countries offering PDC universally and free of charge, adherence remains below policy targets [15]. Several factors can contribute to this, including a lack of awareness about the relevance of dental care during pregnancy [16], dental fear/anxiety and misbelief about dental care [17,18], limited access, a shortage of trained professionals, and insufficient referrals by healthcare teams [19]. Moreover, unprepared professionals may neglect or postpone dental care during pregnancy, undermining the effectiveness of PDC [19].

Given this scenario, strengthening interdisciplinary and educational strategies is essential to improve PDC coverage and reduce maternal–infant risks related to oral health. Integrating dental services into prenatal care represents a critical step in advancing maternal and child health promotion. However, challenges remain in ensuring broad access and adherence to dental treatment. To better understand the factors that may influence this access and adherence, this study aimed to assess the perception of pregnant women undergoing prenatal care regarding oral health and the use of dental services during pregnancy, in public health centers of a Brazilian city.

## 2. Materials and Methods

This report followed the COREQ checklist [20].

### 2.1. Study Design

This qualitative interview-based study is part of a mixed-method research project conducted in collaboration with the Professional Master of Dental Public Health Programme at the Universidade Federal de Minas Gerais (UFMG) and the Brazilian Unified Health System (SUS).

### 2.2. Methodological Orientation and Theory

The study was grounded in theoretical frameworks related to PDC and the understanding of factors influencing its implementation [1,2,6,14,16,18,19].

### 2.3. Setting

The study was conducted in Mariana, a small city in Minas Gerais State, Brazil, with approximately 61,387 inhabitants and 35 SUS-affiliated healthcare facilities, of which 12 provide dental services.

### 2.4. Participant Selection

Participants included pregnant women receiving prenatal care in Mariana’s public health service. Pregnant women were identified through the information system of the Municipal Health Department. Two groups were formed: those who received dental care and those who did not. Women in the latter group either had not been referred for dental care, were unaware of the service, or declined it despite knowing it was available. Using convenience sampling, participants were selected from each health district of the city. Eligible women were aged 18–40, resided in Mariana or its districts, had low to middle income, and at least 8 years of education.

Initially, 10 women with and 10 without dental care were selected, expecting data saturation with 20 interviews [21].

### 2.5. Research Team

The research team underwent training to standardize data collection. Interviews were conducted by two researchers (N.C.F.C. and V.M.G.P.B.), both supervised by an experienced qualitative research professional. V.M.G.P.B. is a public health-trained dentist and local oral health coordinator; N.C.F.C. is a final-year dental student. Researchers had no prior relationship with participants and were trained to minimize personal bias during interviews. At the initial contact with participants, the researchers explained the purpose and motivations of the study. Both interviewers had a strong connection to the topic—one worked as a dentist within the Family Health Strategy and contributed to the organization of the city’s prenatal dental care service. Despite their active involvement and advocacy for the importance of dental care during pregnancy, both were thoroughly trained to ensure that their personal convictions did not influence their relationship with the participants or interfere with the interview process.

### 2.6. Pilot Study

A pilot study was conducted in Mariana to test the interview guide, involving pregnant women from the same population targeted in the main study and using the same methodological approach. The guide was applied until data saturation was reached. As no adjustments were deemed necessary, the participants from the pilot study were included in the final sample.

### 2.7. Data Collection

Data were collected through in-depth, individual interviews conducted with pregnant women receiving prenatal care through SUS in 2024. Each participant was interviewed by a single, designated interviewer using a semi-structured interview guide. Two interview guides were developed, based on the theoretical framework of this study: one for pregnant women who received dental treatment and another for those who did not. The guides were structured into three parts: the first included questions related to personal characteristics, such as age, education level, income, race, etc. The second part addressed the woman’s experience with pregnancy, such as her perception of physiological changes. Finally, the third part included questions about dental treatment, the experience (or lack thereof) with dental care, referral to the municipal oral health service, as well as barriers and facilitators. Each guide contained an average of 25 questions. All interviews were audio-recorded with the participants’ consent and later transcribed [21].

Participants were approached either on the day of their medical or nursing prenatal appointments or during group sessions held for pregnant women, typically in the waiting area. Interviews took place in private rooms within the health centers, ensuring comfort and confidentiality for the participants. Each interview lasted approximately 10 to 15 min and explored topics such as perceptions of oral health during pregnancy, perceived need for dental treatment, personal experiences with dental services, and the support received from healthcare professionals and family members. No participants declined to take part in the study. Field notes were taken during the interviews. No repeat interviews were conducted. Due to limited access and the time-sensitive nature of prenatal care, the transcripts were not returned to participants for comment or correction.

### 2.8. Data Analysis

The interviews were transcribed using Amberscript^®^ software, version 2024, and stored in a digital database. The interview excerpts were translated from Portuguese by one of the authors of this study, who also participated in the interview process and had experience with the English language. The translation was validated by all authors to ensure it accurately reflected the local research context. It was not possible to present the English translations of the interviews to the study participants, as they were not fluent in English. The material was read and reread in an inductive process aimed at identifying emerging themes, emotions, and unexpected perceptions. This iterative reading allowed for a comprehensive understanding of the data and the organization of content into thematic units [21]. The analytical process commenced once the entire set of interviews had been finalized. The content analysis method proposed by Graneheim and Lundman [22] was used to guide the analytical process. This method outlines each analytical step in detail and incorporates key research concepts, offering a structured approach to qualitative data interpretation. It also provides strategies to ensure the reliability of the findings, including credibility, dependability, and transferability throughout the research process [22]. The results were then grouped into themes and categories, followed by a reflective and critical interpretation of the findings.

### 2.9. Ethical Approval

This study complied with all ethical and legal precepts governing research involving human beings, such as the Declaration of Helsinki and Brazilian Resolution n. 466/12 of Health National Counsil. The research protocol was approved by the Ethics Committee of Universidade Federal de Minas Gerais (UFMG) (Project identification code CAAE 74191023.5.0000.5149) and Informed Consent was obtained from all participants.

## 3. Results

A total of 25 pregnant women participated in this study. Some volunteered to take part after seeing their prenatal care peers being interviewed. Although data saturation had already been reached with 20 participants, the additional contributions helped to validate the meanings expressed by the women and to deepen the understanding of the issue under investigation.

The pregnant women’s age ranged from 19 to 39 years. Among them, 84% had more than 8 years of study and 76% a family income above one minimum wage ($304.48 USD at the time of data collection). A total of 80 percent were black or brown women. The gestational age ranged from 10 to 40 weeks, while 68% were in their second pregnancy or more. Among the participants, 40% had visited the dentist during the current pregnancy. There was no difference between the group of pregnant women with and without dental care, except for gestational age (Table 1).

The main findings are presented based on the themes and categories identified through content analysis of the interviews. Representative statements from the participants illustrate their perceptions and experiences regarding dental care during pregnancy, highlighting both the barriers and facilitators to accessing oral health services. The following section outlines the themes and their corresponding categories identified in the study.

### 3.1. Barriers to Accessing Dental Care

The participants identified several barriers that hindered access to dental care during pregnancy. These included health-related conditions that limited the use of services, such as fear of complications for the baby, as well as organizational issues like difficulty obtaining appointments. Additional barriers included misinformation, myths and beliefs about the risks of dental treatment during pregnancy, dental anxiety, and a preference for private care, either due to dissatisfaction with the public service or a desire to continue treatment with a trusted private provider (Table 2).

### 3.2. Changes During Pregnancy

Pregnant women reported undergoing various changes during pregnancy, including physiological, pathological, and emotional changes, as well as feelings related to the discovery of the pregnancy, which ranged from joy to apprehension due to possible complications. These changes also influenced their habits and sense of responsibility, requiring adjustments to daily routines, such as dietary care and increased attention to oral health. Finally, the women highlighted the impact of pregnancy on their quality of life, mentioning the need to adapt their daily activities (Table 3).

### 3.3. Oral Health

Pregnant women’s perceptions of oral health during pregnancy were varied, including concerns about oral changes during this period. Some women reported positive changes in their oral care habits, influenced by dental prenatal care, while others discussed precautions they took related to the baby’s health, such as avoiding certain dental treatments. Additionally, experiences with dental prenatal care were discussed, highlighting the importance of guidance received and education about oral health (Table 4).

### 3.4. Facilitators of Dental Care Access

Several factors contributed to facilitating access to dental care during pregnancy. The participants highlighted the implementation of SUS principles, such as free care and service availability, as well as health education initiatives that helped clarify doubts and dispel myths. Effective scheduling strategies and support from social and community networks were also mentioned as important elements in encouraging adherence to dental care (Table 5).

The humanized care provided by the health team and the satisfaction with the dental team were also points mentioned by the pregnant women. Their statements emphasized the trust in the care received, built through the attention and empathy of the dentists, and the importance of a welcoming communication, which helped reduce fear and anxiety. Satisfaction with the dental team reflected the perception that the care received was essential for both oral and overall health during pregnancy.

## 4. Discussion

The results of this study reveal the various perceptions of pregnant women regarding dental care during pregnancy, highlighting barriers, facilitators, organizational aspects, and the impact of gestational changes on their general and oral health. There were no differences in responses between primiparous and multiparous women. There were also no differences between women with and without dental care regarding their perceptions and fears, with the exception of prenatal dental care experience. When comparing the findings with the existing literature, important implications emerge for the organization of health services, especially within the scope of prenatal dental care.

### 4.1. Barriers to Dental Care

Pregnant women face a series of barriers to accessing dental care during pregnancy, often related to cultural, emotional, and structural issues. Popular beliefs and lack of clear information can heighten fear and lead to the avoidance of dental services. The fear that dental procedures could harm the baby and the belief that pregnancy causes tooth loss because the fetus “takes” calcium from the mother are myths that still keep many pregnant women away from dental care [16]. These beliefs reinforce the need for educational strategies that help clarify these issues and provide greater assurance for pregnant women to seek necessary care during this period.

In addition, organizational issues such as difficulty scheduling appointments and a lack of available times compatible with the pregnant women’s routines were pointed out in the interviews. These factors also align with findings in the literature, highlighting the need to make SUS services more flexible to improve access [18,19,23].

Another observed aspect was the choice of private care, often motivated by convenience and the perception of easier access. This behavior also reflects a lack of awareness about the dental services available through SUS.

### 4.2. Transformation During Pregnancy

The changes experienced by women during pregnancy involve physiological, emotional, and behavioral aspects, directly influencing their quality of life and adherence to health practices. Emotional changes—common during pregnancy—can intensify feelings of vulnerability and affect overall well-being, as noted in several studies that reported the impact of pregnancy on women’s emotional health [24].

In addition, the emotional impact of discovering the pregnancy was emphasized by several participants, especially in high-risk pregnancies, often generating mixed feelings of joy and apprehension. Interestingly, some participants reported dealing with these feelings in different ways: while some sought family or professional support, others preferred not to share their concerns, which intensified the emotional burden. These reports support findings by Figueiredo et al. [1], who describe early pregnancy as a time of increased responsibility, but also of doubts and anxieties regarding maternal and fetal health.

Changes in habits and routines were also mentioned, particularly related to diet, hygiene, and oral healthcare. These adjustments, however, were often limited by emotional and cultural barriers [25]. Behavioral changes are more effective when pregnant women receive both educational and emotional support during prenatal care [26].

Finally, gestational changes directly affected participants’ ability to cope with everyday demands [27]. For some, the burden of physical and emotional changes disrupted their routines and affected their quality of life [28]. Studies such as that of Hodgson and collaborators [29]—the first Canadian study to compare outcomes in pregnant women receiving interprofessional group care versus individual care—emphasize the importance of interdisciplinary support that addresses both clinical and emotional aspects of pregnancy, promoting comprehensive access to needed care.

### 4.3. Oral Health Condition During Pregnancy

Research indicates that pregnant women with poor oral health at the beginning of pregnancy are at higher risk for obstetric complications [8,9]. A meta-analysis of randomized controlled trials evaluating the efficacy of scaling and root planing in reducing the preterm-birth and low-birth-weight risks showed combined risk ratios of 0.66 (95% CI = 0.54, 0.80) when analysis was restricted to women at excess prematurity risk [30]. These outcomes are associated with an exacerbated inflammatory state, which triggers intense immune responses in the maternal body [31]. In addition, the physiological changes characteristic of pregnancy, including altered immune responsiveness and inflammatory activity [32], may worsen pre-existing oral conditions such as gingivitis and periodontal disease [2], and contribute to the development of other issues like aphthous ulcers, dental erosion, halitosis and dry mouth [5]. These relationships between oral health and pregnancy were acknowledged by participants at various points throughout the interviews. One woman shared, *“I have some tooth sensitivity, in one tooth, this specific one here. (…) I bought Sensodyne (laughs) (…) it hasn’t improved, but it’s helping.”* This statement highlights oral sensitivity as a common change during pregnancy [33].

It was also possible to observe the impact that participation in prenatal dental care had on the hygiene practices of pregnant women. One participant, for example, said, *“I’ve started brushing more. (…) Sometimes, in the rush, we brush once or at most twice, right? Now I’m being more strict and brushing three times.”* This statement demonstrates how dental follow-up can lead to concrete changes in oral hygiene practices, reinforcing the importance of integrating educational initiatives into prenatal care [26,34].

### 4.4. Facilitators of Prenatal Dental Care

Despite the barriers highlighted during the interviews, the pregnant women identified important facilitators that supported their adherence to prenatal dental care. These facilitators reflect both individual characteristics and organizational aspects of the healthcare system. Identifying and understanding these factors is essential for strengthening oral health promotion strategies during pregnancy.

The Brazilian Unified Health System (SUS) plays a central role in providing dental care to pregnant women, promoting access through primary healthcare and the regional organization of services. The National Oral Health Policy emphasizes the importance of dental care during pregnancy, integrating it into maternal and child health and recognizing its relevance for the health of both mother and baby [14]. The Brazilian guideline for clinical dental practice during pregnancy in primary healthcare recommends that all pregnant women should have at least one dental appointment during pregnancy in public dental services [35]. This institutional support was perceived by the participants as a key facilitator in accessing oral healthcare, as expressed in one account:


*“(…) I found it much easier, I thought that it was just something minor. SUS actually covers a lot more than my health insurance plan, to be honest (…).”*


In line with SUS guidelines, pregnant women are considered a priority within health services [14]. This differentiated care is often recognized by women themselves as a factor that facilitates access to the necessary services [18,19,36]. Participants highlighted this prioritization through actions such as immediate appointment scheduling at the beginning of prenatal care, proactive outreach via telehealth, and streamlined access due to their pregnancy status. Additionally, the practice of coordinating medical and dental appointments (“joint scheduling”) was also cited as an important facilitator. As one participant explained, *“(…) I didn’t even have to look for it, they referred me right away when I found out I was pregnant (…) the nurse have already referred me (…).”*

During pregnancy, women are often more receptive to health information related to themselves and their babies. This openness allows them, when well-guided and motivated, to adopt health-promoting behaviors, reinforcing the role of prenatal dental care [26]. In addition, women who receive proper guidance during and after pregnancy tend to demonstrate greater knowledge and commitment to their family’s oral health, showing that health promotion efforts during this period are important for establishing habits that have a positive impact on the early years of a child’s life [34]. A Cochrane systematic review of randomized controlled trials showed evidence that providing advice on diet and feeding to pregnant women reduces the risk of early childhood caries in the offspring [37]. In the same way, a Brazilian cohort study with 2287 mother–child dyads demonstrated that children whose mothers visited the dentist during pregnancy or within the last year for preventive reasons or curative reasons were more likely to have visited the dentist during the first year of life compared to those whose mothers had not used dental services during this period [38]. Early and regular dental visits in childhood are strongly linked to better quality of life in later life [39]. Therefore, the efforts on health promotion through prenatal dental care can have lasting positive effects on an individual’s life.

Social and community networks also play an important role in encouraging pregnant women to participate in prenatal dental care. Family support—especially from mothers, sisters, and partners—was described as a form of emotional support and motivation. In addition, shared experiences with friends and neighbors reinforced the importance of oral care during pregnancy. Studies such as those by Rocha et al. [40] also emphasize how support networks can help facilitate access and adherence to healthcare services, helping to overcome emotional and cultural barriers. On the other hand, if the culture and popular knowledge within these networks are influenced by misconceptions and prejudices about dental care during pregnancy, women may avoid treatment [18]. Therefore, health promotion efforts aimed at pregnant women should be aligned with principles of community orientation, family focus, and cultural competence, valuing the local knowledge within the community [41].

The care provided by the healthcare team is another relevant facilitator. The close relationship with the healthcare team serves as an important foundation for pregnant women, who are often experiencing new situations that bring both physiological changes and shifts in daily routines. When there is a relationship of trust, healthcare professionals play a highly influential role in the pregnant woman’s life, often being responsible for encouraging participation in prenatal dental care [42], as evidenced in the interviews. Interestingly, pregnant women who are informed by dentists about the importance of dental treatment during pregnancy are twice as likely to have a dental appointment; however, those who are advised and referred to the dentist by other prenatal provider are four times more likely to visit a dentist, demonstrating the enormous influence that these professionals have on the pregnant woman [43]. It is no coincidence that the pregnant women in our study reiterated in their interviews that if the healthcare team, specially the physician, had informed them, they would have visited the dentist sooner, thus highlighting the bonding with the healthcare team and the trust they place in these professionals.

Humanized care, in which healthcare professionals treat each woman individually, with empathy and respect, was highly valued by the participants. One pregnant woman shared, *“Oh, I felt really welcomed. She treats us very well. She talks to us respectfully, explains things (…) she also talks about the baby, all those things.”* This statement highlights the importance of attentive listening and emotional support provided by the healthcare team, key elements for making pregnant women feel cared for and safe throughout prenatal care. Studies conducted in Latin America on humanized prenatal care recommend qualifying health professionals to strengthen humanized prenatal care, considering pregnant women as protagonists of the process, dispelling their doubts and concerns, and provide evidence on the relationship between humanized prenatal care and the reduction in maternal morbidity and mortality [44].

Another important point raised by the participants was how satisfaction with the care received directly impacts their adherence to health recommendations, demonstrating how trust in the healthcare team can lead women to follow professional guidance more effectively, supporting positive lifestyle changes. On the other hand, negative experiences with healthcare professionals can generate distrust and insecurity, making it more difficult to engage with prenatal dental care.

The care provided by the health team can significantly shape the experiences of pregnant women positively, by facilitating access and adherence to prenatal dental care, or negatively, by creating barriers and feelings of insecurity. Several studies have emphasized the importance of a humanized approach and effective communication to strengthen the bond between pregnant women and health services, promoting more effective follow-up [18,23,40,44].

This study presents both limitations and strengths. Among the limitations are the specificities of the Brazilian context and the municipality where the data were collected. The subjectivity of the researchers and their unconscious biases may affect the analysis of qualitative data. However, this is also a limitation of the method adopted. Another point that should be mentioned was the difficulty in accessing more pregnant women who received prenatal care at distant health units in rural areas. As for the strengths, the study contributes to the scientific framework on dental care for pregnant women by presenting original data that deeply analyze the different perceptions of pregnant women with and without access to dental treatment. It contributes to improving understanding of the challenges pregnant women face in accessing care, while also offering insights for enhancing the organization and provision of dental services during prenatal care. The Brazilian experience may also be valuable to other countries seeking to organize the provision of prenatal dental care through a health system, guided by a primary healthcare approach.

## 5. Conclusions

Prenatal dental care is an essential component of healthcare during pregnancy, contributing to a safer and healthier experience for the mother. However, there are barriers that limit access to this service, which involve contextual, emotional, and cultural factors. The lack of awareness about the need for dental care along with misinformation regarding the safety of undergoing dental procedures during pregnancy—often influenced by myths, beliefs, and taboos—were identified as the main barriers.

On the other hand, the study’s findings revealed several facilitators that support the adherence to prenatal dental care in Brazil, particularly the availability of care through the SUS, which plays a crucial role in providing these services. Nonetheless, the system still faces organizational weaknesses that need to be addressed. Emphasizing health education and implementing strategies that simplify dental appointment scheduling for pregnant women are essential measures to ensure better adherence and improve the effectiveness of this service.

Future studies should consider expanding the sample to include diverse populations across different geographic and socioeconomic contexts, which could enhance the generalizability of the findings. Additional qualitative data collection strategies should also be considered, such as focus groups. An important aspect to be further explored is the potential for dialogue on the barriers and facilitators of PDC among various stakeholders, including healthcare professionals from different categories and pregnant women. In this regard, research needs to advance in proposing strategies to address the problem by establishing interventions that can contribute to the promotion of comprehensive and equitable oral healthcare for pregnant women.

## Figures and Tables

**Table 1 ijerph-22-01211-t001:** Sociodemographic characteristics of participants.

Variable		Pregnant Women with PDC	Pregnant Women Without PDC
Age	≤19 years	1 (09.1%)	0
	20–29 years	6 (54.5%)	10 (71.4%)
	30–39 years	4 (36.4%)	4 (28.6%)
	40 or more years	0	0
Race	White	2 (18.2%)	1 (07.1%)
	Brown	6 (54.5%)	8 (57.2%)
	Black	2 (18.2%)	4 (28.6%)
	Asian	0	1 (07.1%)
	Do not know	1 (09.1%)	0
Education level	<8 years of study	1 (09.1%)	1 (07.1%)
	≥8 years of study	10 (90.9%)	13 (92.9%)
Family income	<1 minimum wage *	2 (18.2%)	1 (07.1%)
	≥1 minimum wage *	8 (72.7%)	12 (85.8%)
	Do not know	1 (09.1%)	1 (07.1%)
Primiparous	Yes	4 (36.4%)	4 (28.6%)
	No	7 (63.6%)	10 (71.4%)
Gestational age	1–13 weeks	3 (27.3%)	3 (21.4%)
	14–26 weeks	2 (18.2%)	8 (57.2%)
	27–40 weeks	6 (54.5%)	3 (21.4%)

Legend: * minimum wage = $304.48 USD at the time of data collection.

**Table 2 ijerph-22-01211-t002:** Pregnant women’s reports on barriers to dental care.

Barriers to Dental Care	
Health conditions during pregnancy that limit the use of dental service	*“The day the girl called me to remind me about my dentist appointment, I was feeling really sick, and I told her I wasn’t going because I was vomiting all the time, so I couldn’t go out, right, vomiting?”* [Pregnant woman 07]
Organizational problems in the health service	*“The reason is that they don’t call us, they only advertise it, nothing else. (…) They put up flyers, but that’s no good. Look at the number of pregnant women there today. There must be about 20, and of those, only two must have gotten an appointment. They have to advertise and also promote the service.”* [Pregnant woman 08] *“It takes time to get an appointment.”* [Pregnant woman 16]
Lack of information	*“I didn’t know. My doctor didn’t mention anything. I had no information. No one told me to go to the [dentist] or anything.”* [Pregnant woman 04] *“A lot of people don’t know.”* [Pregnant woman 05]
Myths and beliefs	*“The dentist said he wouldn’t do any dental work on me. He said it was only after the baby was born, that it was dangerous. I, out of fear, didn’t seek another one either. From what they told me, it could even cause me to lose the baby. I got really scared, right?”* [Pregnant woman 12] *“I think that during pregnancy, I don’t know if it’s okay to have procedures (…) Even getting cavities, people say pregnancy causes more cavities. I don’t know if that’s true or not.”* [Pregnant woman 20] *“I didn’t get an X-ray, I was kind of scared.”* [Pregnant woman 10] *“He [the dentist] said I didn’t need to do anything now. He said that now it’s no use doing other stuff, because I can’t, because of the anesthesia.”* [Pregnant woman 06]
Anxiety toward dental treatment	*“I’m terrified! I was traumatized, ever since I was little. (…) That drill (…) Oh my God! My husband went, I didn’t. Even the dentist said I could get braces (…), I can put them on (…). As for the rest, during pregnancy, no. (…) Right now, I don’t want to go either. I’m too nervous. I’m terrified.”* [Pregnant woman 06] *“During my son’s pregnancy, I suffered a lot (…) I kept taking those needles, everything scared me.”* [Pregnant woman 15]
Preference for private dental care	*“I already go to a private dentist, so for me public care wasn’t feasible.”* [Pregnant woman 13]

**Table 3 ijerph-22-01211-t003:** Pregnant women’s accounts of changes during pregnancy.

Changes During Pregnancy	
Physiological, pathological, or emotional changes during pregnancy	*“My mood changed a lot. I cry over anything (…). I’ve gained a lot of weight already.”* [Pregnant woman 07] *“(…) I felt really sick, lost five kilos. (…) I was nauseous day and night. I’d sleep and wake up feeling nauseous. People would ask: is there something that’s making you sick? Oh, even the air is making me sick. I can’t breathe because it makes me want to vomit (…)”* [Pregnant woman 19]
Feelings about discovering the pregnancy	*“I was planning it, right? I was happy. Joyful, right? A little scared, anxious, but I think that’s normal too.”* [Pregnant woman 12]
Changes in habits and sense of responsibility	*“I became more responsible. I started thinking more about the future, because before it was just the two of us (me and my husband). Now there’s a little one who needs us. So that’s it (…).”* [Pregnant woman 12] *“I stopped drinking, I stopped smoking. I stay more at home now. (…) In my other pregnancy I was working, not in this one.”* [Pregnant woman 13]
Impact of pregnancy on daily routine and quality of life	*“Ah, I changed a lot (…). Food-wise (…) I used to eat anything, rode a motorcycle (…). Now I’m more calm, I eat better.”* [Pregnant woman 23] *“(…) I didn’t change (…) pretty much nothing. I can do everything normally. There’s no difference.”* [Pregnant woman 5] *“The routine slowed down (…). We can’t do the same things we used to anymore (…).”* [Pregnant woman 10] *“The change is that I had to work a lot less (…).”* [Pregnant woman 21]

**Table 4 ijerph-22-01211-t004:** Pregnant women’s accounts about oral health during pregnancy.

Oral Health	
Oral changes during pregnancy	*“I felt bleeding. When I brush my teeth, it bleeds a little every time. I’ve looked it up a lot and saw that it’s normal to have bleeding and gum sensitivity during pregnancy.”* *“Dental pain, (…) I couldn’t take it, (…) I felt really sick. (…) I thought I was going to faint from so much pain (…)”* [Pregnant woman 12] *“I have a lot of tooth sensitivity.”* [Pregnant woman 4] *“(…) I started noticing a bitter taste in my mouth. (…) My throat was bitter.”* [Pregnant woman 14] *“Everything was normal.”* [Pregnant woman 24]
Dental care for the baby	[Would you take the baby to a dental appointment?]. *“No, there’s no need. (…) If I was told I needed to, I would take them. But up until now, I think I wouldn’t take them. (…) I’d think they don’t have teeth yet.”* [Pregnant woman 04]
Habit changes due to PDC	*“I brush more often now. (…) Sometimes, in the rush, we brush once or maybe twice, right? Now I’m brushing more carefully, three times.”* [Pregnant woman 02] *“(…) you have to brush more often, floss more often, even to avoid gingivitis, tartar buildup, because with the nausea we end up building up more and more (…)”* [Pregnant woman 10]
Perception of oral health	*“It’s good to take care of your teeth, of your mouth too, (…) if you don’t, it can cause illness for the baby during pregnancy. (…) A pregnant woman has to take urgent care because of the baby.”* [Pregnant woman 14]
Experiences with PDC	*“It was great. The dentist gave a talk, explained about the mother’s and baby’s oral health, and then she also did an evaluation. Now I have a return appointment tomorrow for a procedure.”* [Pregnant woman 17] *“I had treatment, everything was nice, I was able to get anesthesia. I even got treatment because I had a tooth that was hurting.”* [Pregnant woman 04]

**Table 5 ijerph-22-01211-t005:** Pregnant women’s reports on facilitators of dental care access.

Facilitators for Dental Care	
Materialization of SUS doctrinal principles	*“It’s very good to be through SUS (…). If I hadn’t gone through SUS, I wouldn’t be taking the injections I need to take, I’d have to pay (…). I have nothing to complain about.”* [Pregnant woman 06] *“(…) I thought it was much easier, I thought it (…) was something you only did for little kids. SUS covers more than my health plan, to be honest (…).”* [Pregnant woman 21] *“Since it’s through SUS, it’s always good, it’s already within our reach.”* [Pregnant woman 17] *“The dentist at the public health unit] asked me if I was interested in continuing later on (…). I said of course I was (…). It’s free, of course (laughs)!”* [Pregnant woman 21]
Health education	*“Doctors should inform (…). No one links dentists with pregnancy (…). So if doctors informed, it would be good (…) most have dental problems during pregnancy. I don’t know what happens (…) but you go years without going and then you get pregnant and things start.”* [Pregnant woman 04] *“(…) the obstetrician called me to talk. I think that helps a lot (…). Community health agents also going around and informing (…) I think it helps a lot to spread the information that there is prenatal dental care.”* [Pregnant woman 09] *“(…) there should be more campaigns, more information (…). Some people don’t have much information (…). They don’t know they’re entitled to it. (…) There should be more access to this on TV, in schools, informing the children and they would already bring a note home to their parents. Sometimes, the mothers are pregnant and they’ll already be informed about everything.”* [Pregnant woman 10] *“(…) Announce it (…), go, call, speak on the radio, (…) today we have all kinds of communication. There’s Facebook, (…) radio, (…) practically everything, (…) social media. We can’t say it’s just up to the dentist. Sometimes we ourselves, the population, (…) don’t want to know (…).”* [Pregnant woman 14] *“(…) if there were more talks like this (…), every month. I think that would encourage a lot.”* [Pregnant woman 03] *“I think what hurts us the most is fear. So the fear of that anesthesia, everything hurts more. But the dentist talking and everything. I think that helps a lot.”* [Pregnant woman 07] *“Maybe promote more, (…), even during prenatal consultations, say that you can go, that there’s no problem, if needed.”* [Pregnant woman 19]
Strategies for scheduling pregnant women’s dental appointments	*“It was this part of being referred directly (…) the process was quicker. I didn’t have to wait in line, it was already (…) prioritized for pregnant women. It’s much easier to be referred directly.”* [Pregnant woman 11] *“(…) What helped was the nurse referring me.”* [Pregnant woman 07]
Social and community networks	*“My sisters recently had a baby and participated in the prenatal dental care. She already knew (…) that I was going to participate too. And I have friends on my neighborhood who also participated.”* [Pregnant woman 21]
Humanization in care	*“Oh, I felt really welcomed. She treats us very well. She talks to us respectfully, explains things. She also talks about the baby, those kinds of things, everything.”* [Pregnant woman 02]
Satisfaction with care	*“Every time I go, I’m well taken care of (…).”* [Pregnant woman 06] *“I liked the care.”* [Pregnant woman 23]

## Data Availability

The data presented in this study are available on request from the corresponding author due to ethical restrictions imposed by the public health service from where data were collected.

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
