# Peer review of "Changes, Desire, Fear and Beliefs: Women’s Feelings and Perceptions About Dental Care During Pregnancy"

_ijerph, 2025, doi:10.3390/ijerph22081211_

Round 1
Reviewer 1 Report
Comments and Suggestions for Authors
With interest I’ve read the paper “Changes, Desire, Fear and Beliefs: women's feelings and perceptions about dental care during pregnancy”. This paper reports on the qualitative phase of a mixed-method study conducted with 25 pregnant women receiving prenatal care in the Brazilian Unified Health System. It describes emotional, cultural, and contextual aspects and how they influence the use of dental services during pregnancy.
The topic is important and relevant. The study is well-planned and well-written and followed the COREQ checklist. However, there are some revisions to be made before publishing.
In the introduction, the phrase «Despite Brazil being the only country offering PDC universally and free of charge, adherence remains below policy targets» needs further comments, as many countries offer free and prioritized dental care in pregnancy (https://juniperpublishers.com/jgwh/pdf/JGWH.MS.ID.555958.pdf
https://pmc.ncbi.nlm.nih.gov/articles/PMC4722780/
https://pmc.ncbi.nlm.nih.gov/articles/PMC10740224/, etc.).
Methods. More information is needed on the development of the interview guide and its content.
As far as I understand, the interview was in Portuguese? How were the phrases presented in the article translated? How was the correctness of the translation ensured?
Results. When describing the population of the study it is important to give descriptive information on the two groups separately (such as age, education, income, etc); it would be illustrative to present this data as a table.
Were there any differences between the groups regarding their perceptions and fears?
Author Response
With interest I’ve read the paper “Changes, Desire, Fear and Beliefs: women's feelings and perceptions about dental care during pregnancy”. This paper reports on the qualitative phase of a mixed-method study conducted with 25 pregnant women receiving prenatal care in the Brazilian Unified Health System. It describes emotional, cultural, and contextual aspects and how they influence the use of dental services during pregnancy.
The topic is important and relevant. The study is well-planned and well-written and followed the COREQ checklist. However, there are some revisions to be made before publishing.
Authors' response to the journal reviewer: Dear reviewer, your contributions have been invaluable for refining our manuscript. Thank you for taking the time to evaluate our paper. We have taken all your suggestions into account and made the necessary revisions. Below, we respond point by point to all the questions raised.
In the introduction, the phrase «Despite Brazil being the only country offering PDC universally and free of charge, adherence remains below policy targets» needs further comments, as many countries offer free and prioritized dental care in pregnancy (https://juniperpublishers.com/jgwh/pdf/JGWH.MS.ID.555958.pdf
https://pmc.ncbi.nlm.nih.gov/articles/PMC4722780/
https://pmc.ncbi.nlm.nih.gov/articles/PMC10740224/, etc.).
Authors' response to the journal reviewer: Dear reviewer, thank you for your observation. Brazil offers prenatal dental care to all pregnant women in the country, including dental treatments at the primary, secondary, and tertiary care levels. We recognize that there are few countries that provide dental care to pregnant women in this way at a national level; however, Brazil is not the only one, as evidenced by the articles shared by this reviewer. Therefore, we have rephrased our sentence as follows: “Despite Brazil being one of the few countries offering PDC universally and free of charge, adherence remains below policy targets”
Methods. More information is needed on the development of the interview guide and its content.
Authors' response to the journal reviewer: Dear reviewer, we agree with your comments and have included the following additional information about the development of the interview guide and its content on the manuscript: “Two interview guides were developed, based on the theoretical framework of this study: one for pregnant women who received dental treatment and another for those who did not. The guides were structured into three parts: the first included questions related to personal characteristics, such as age, education level, income, race, etc. The second part addressed the woman's experience with pregnancy, such as her perception of physiological changes. Finally, the third part included questions about dental treatment, the experience (or not) with dental care, referral to the municipal oral health service, as well as barriers and facilitators. Each guide contained an average of 25 questions”.
As far as I understand, the interview was in Portuguese? How were the phrases presented in the article translated? How was the correctness of the translation ensured?
Authors' response to the journal reviewer: Yes, it was in Portuguese. The interview excerpts were translated by one of the authors of this study, who also participated in the interview process and had experience with the English language. The translation was validated by all authors to ensure it accurately reflected the local research context. It was not possible to present the English translations of the interviews to the study participants, as they were not fluent in English. We had added this information in the manuscript.
Results. When describing the population of the study it is important to give descriptive information on the two groups separately (such as age, education, income, etc); it would be illustrative to present this data as a table.
Authors' response to the journal reviewer: Dear reviewer, thank you for your suggestion. We included the table with participants’ characteristics, with the two groups separately.
Were there any differences between the groups regarding their perceptions and fears?
Authors' response to the journal reviewer: No, there isn´t. The difference was only regarding the experience of dental treatment during pregnancy. We included this information in discussion section.
Reviewer 2 Report
Comments and Suggestions for Authors
This study addresses pregnant women’s views and behaviors concerning oral health and dental care access during pregnancy under the Brazilian SUS model. The subject matter is highly pertinent and of contemporary relevance, given the established importance of maintaining oral health during pregnancy and its potential implications for both maternal and perinatal outcomes. This study offers a meaningful contribution by elucidating the cultural, emotional, and systemic factors that influence the utilization of dental services among pregnant women.
My comments are below.
1. The introduction is very well written.
2. The discussion section is overly extensive and should be shortened. I believe that the direct quotations of pregnant women's statements should be removed from the discussion.
3. The authors should specify, if possible, whether they observed any differences in responses between primiparous and multiparous women.
4. Mentioning additional limitations of the study would strengthen the manuscript.
5. Providing more detailed suggestions for future research, including what elements should be considered in a quantitative analysis, would be beneficial.
Author Response
This study addresses pregnant women’s views and behaviors concerning oral health and dental care access during pregnancy under the Brazilian SUS model. The subject matter is highly pertinent and of contemporary relevance, given the established importance of maintaining oral health during pregnancy and its potential implications for both maternal and perinatal outcomes. This study offers a meaningful contribution by elucidating the cultural, emotional, and systemic factors that influence the utilization of dental services among pregnant women.
My comments are below.
- The introduction is very well written.
Authors' response to the journal reviewer: Dear Reviewer, thank you for your valuable contributions. Your suggestions have significantly enhanced the quality of our manuscript.
- The discussion section is overly extensive and should be shortened. I believe that the direct quotations of pregnant women's statements should be removed from the discussion.
Authors' response to the journal reviewer: Dear Reviewer, we have eliminated the majority of direct quotations of pregnant women's statements, retaining only those that are essential to the discussion section.
- The authors should specify, if possible, whether they observed any differences in responses between primiparous and multiparous women.
Authors' response to the journal reviewer: There were no differences in responses between primiparous and multiparous women. We have added this information to the text.
- Mentioning additional limitations of the study would strengthen the manuscript.
Authors' response to the journal reviewer: We mentioned additional limitations in the text: “This study presents both limitations and strengths. Among the limitations are the specificities of the Brazilian context and the municipality where the data were collected. There is the subjectivity of the researchers and the unconscious biases that may affect the analysis of qualitative data. However, this is also a characteristic limitation of the method adopted. Another point that should be mentioned was the difficulty in accessing more pregnant women who received prenatal care at distant health units in rural areas”.
- Providing more detailed suggestions for future research, including what elements should be considered in a quantitative analysis, would be beneficial.
Authors' response to the journal reviewer: Dear reviewer, thank your analysis. We included the following sentence in the manuscript’s conclusion: “Future studies should consider expanding the sample to include diverse populations across different geographic and socioeconomic contexts, which could enhance the generalizability of the findings. Additional qualitative data collection strategies should also be considered, such as focus groups. An important aspect to be further explored is the potential for dialogue on the barriers and facilitators of PDC among various stakeholders, including healthcare professionals from different categories and pregnant women. In this regard, research needs to advance in proposing strategies to address the problem by establishing interventions that can contribute to the promotion of comprehensive and equitable oral health care for pregnant women”.